# Evaluating the operational efficiency of NBA teams on franchise value: An assessment of data envelopment analysis

**Philsoo Kim**[1]* , **Sang Hyun Lee**[1] , **JeongJun Moon**[2]

1 Korea Sport Management Research Institute, Seoul, Republic of Korea, 2 Department of Statistics, University of Pittsburgh, Pittsburgh, Pennsylvania, United States of America

☯ These authors contributed equally to this work.

* shscottlee@naver.com

**Data Availability Statement:** All data files are available from the NBA.com database

**Funding:** The author(s) received no specific funding for this work.

## Abstract

The main purpose of this research is to empirically analyze the determinants of organizational performance using National Basketball Association (NBA) team data. Based on the resource-based theory of the firm, prior studies posit that operational efficiency encompasses the ability of professional sports teams to translate their resources into creating organizational performance. The contention is that NBA teams enhance organizational performance in the market when possessing valuable, rare, inimitable, and non-substitutable resources and capabilities. In this sense, the operational efficiencies of NBA teams align with the concept of core competence, enabling teams to achieve competitive advantages through superior performance. The exploration of the level of operating efficiency in NBA teams and its role in organizational performance is beyond essential. This study conceptualizes operating efficiency as the degree of competence exhibited by professional sports teams, drawing on comprehensive game-related statistics and financial performance data derived from human assets and team budgets. To bridge theory and empirical investigation, data spanning six seasons (2015–2016 to 2020–2021) for all 30 NBA teams were collected. The results reveal that 29 out of 180 decision-making units exhibit outstanding organizational efficiency, significantly contributing to franchise value.

## Introduction

Within the context of professional sports, the term "operational efficiency" holds significance, representing the output generated based on input [1]. However, in the sports domain, this term is often used interchangeably with "effectiveness." The rationale for this interchangeability lies not only in the common attributes of the terms but also in the unique aspects of professional sports franchises. Despite competing in similar environments, as exemplified by National Basketball Association (NBA) teams in the league, the pursuit of maximizing performance within the constraints of a salary cap necessitates a direct relationship between team efficiency and effectiveness [2–5].

The representation of sports franchises and type of business units are obliged to be efficient seeking not only the goal of profit maximization but also to increase its franchise value [6,7].

**Competing interests:** The authors have declared that no competing interests exist.

One of the key strategies and goals of contemporary professional sports teams is to secure a championship by accumulating regular season and postseason wins and maximize the franchise value This entails strategic investments within budget constraints to acquire quality players and optimize management efficiency. Furthermore, sports teams need to achieve financial return to have core resources to compete with other teams in the league. The presence of excellent players, their performance, and the available capital for recruiting and motivating these players have a positive impact on team wins to a certain extent [8,9].

Despite perennial questions regarding which team is the most efficient in the league [10–12], our focus is on the fundamentals of team-aggregated player performance leading to the level of organizational efficiency. The presence of talented players or financial resources does not always translate into or substitute for operational efficiency of professional sports teams that should be leveraged for franchise value [13,14]. Prior literatures have shed light on estimating the operational efficiency of sports organizations by comparing the performance of sports teams, such as athletic performance, ranking, and winning percentage [15–18]. Other studies have endeavored to assess efficiency by considering financial metrics like sales, revenue, and budget investments [19–21]. However, the context in which a sports organization is efficient should be defined in a comprehensive way to reflect its unique features, which combine both sport-specific and financial aspects of the franchise. This is because the value of a sports franchise should include dual aspects, as they are both sport-specific entities and business units.

The notion of resource-based theory (RBT) provides a clear logic on organizations being capable of seeking sustainable competitive advantage to perform better based on their level of core competencies [22,23]. Sports franchises, accordingly, must possess core competencies characterized as valuable, rare, inimitable, and difficult to substitute. In line with this argument, [24] defined the operating efficiency of a professional sports team as the ability to transfer its resources into winning. Such organizational resources include the budget spent to maintain human capitals and the aggregated capabilities of players into the professional sports franchise. It is critical for a sports franchise to not only possess superior level of human assets that should perform well for the winning of the game, but also to retain capabilities to transfer such resources into operational efficiencies that should maximize franchise value. The value of a sports franchise depends not only on the team's rank in the league, but also on its financial status [12]. This means that sports teams' performance should be evaluated based on both on-field performance and financial performance. Thus, the efficiency of a sports franchise can be comprehensively estimated by considering the budget invested, human assets, and both game and financial performance. Despite this, studies considering on-field and financial performance as output variables, with budgets invested and human assets as input variables, are scarce in the sports domain.

Accordingly, this research aims to estimate the operational efficiency of professional sports organizations using the budget invested in team-aggregated player capabilities as input factors, with game performance and financial gain as output factors. Team operating efficiency is defined as the ability of a professional sports team to effectively utilize its inherent organizational resources to achieve organizational performance. It stands as a key determinant of team value, as teams with high operating efficiency can construct more attractive teams in the market, even under identical conditions. Given that most NBA teams compete under relatively similar circumstances, operating efficiency may play a crucial role in determining organizational performance, even when resources are comparable [25,26]. Consequently, we argue that team operating efficiency is an inherent and unique determinant of organizational performance, serving as an essential organizational resource that enables teams to gain a comparative competitive advantage in the NBA.

In this regard, the objective of this research is to provide a theoretical framework for the comprehensive evaluation of professional sports franchise operational efficiency using NBA team data. This study employs data envelopment analysis (DEA), a widely utilized method in the field of professional sports to compare the efficiency of players or teams relative to their inputs [27–29]. The data used in this study was collected from 6 seasons (2015~2016 to 2020~2021) of NBA game records. Through the application of DEA, we aim to estimate which team is more efficient than others and further examine the impact of team operational efficiency on the increased Forbes franchise value. We believe this study can contribute to the domains of sports analytics by analyzing NBA team operating efficiency. The idea of considering the quality of human resources of a sports team alongside financial investment as input factors and both on-field and financial performance as output factors can provide insight to sports stakeholders on how to assess the resources sports teams possess and utilize them effectively.

## Theoretical background

### Resource-based theory and operating efficiency

This research theoretically applies and extends the resource-based theory to elucidate the impact of operational efficiency on the value of professional sports franchises. Originating in the field of strategic management, RBT identifies organizational characteristics that confer advantages in competitive environments [22,30,31]. Empirically supported over the years, RBT has become integral in sports management literature, explaining determinants of organizational performance for professional sports teams [9,32–37].

RBT posits that organizations with resources possessing intrinsic value, rarity, difficulty of imitation, and non-substitutable attributes can achieve sustainable performance advantages—collectively termed as core competencies. Such organizations can effectively execute strategies, attain objectives, and maintain a competitive advantage in the market [22,30,31]. When applied to professional sports teams, RBT suggests that possessing core competencies not found in other organizations leads to differentiated performance compared to competing teams in the same league [9,35,38]. Building on RBT, two types of core competencies for sports franchises are identified [38–40].

The first resource is human assets, specifically the players, who are crucial as the core products of sports organizations are game performances. Yet, assessing individual player proficiency at the organization level requires considering the aggregated game performance of a team. Another vital resource is the budgets invested in players, i.e., team salaries. Salaries play a fundamental role in attracting, retaining, and motivating valuable human assets. In this context, the total salaries of a team are considered as organizational resources.

RBT posits that organizational resources encompass both tangible and intangible aspects [38,40]. Operational efficiency is viewed as an intangible core competence of sports franchises, providing the ability to gain competitive advantages through superior performance. Efficiency is typically measured as the ratio of inputs to outputs. An efficiency study in the sports domain defined operating efficiency as the ability to transfer sports team resources into winning [24]. Extending this definition, we define operating efficiency as the ability to transfer organizational core competencies into organizational performance.

The choice of input-output variables in efficiency equations depends on the study's purpose. However, previous sports domain studies often overlooked human assets, a critical core competency of sports franchises, by only using financial investment as input variables [27,41–45]. Recognizing human assets and financial resources as critical tangible resources in sports management, we argue that both human capital and total team salaries should be included as input variables when estimating operating efficiency.

According to a recent RBT review paper in the sports domain [38], human assets and financial resources are considered critical tangible resources in sports management. [46] also argue that capital and human assets are the main categories of input variables when estimating the efficiency of organizations. As argued earlier, human capital and budgets invested in sports teams are critical resources, and both need to be considered as input variables when estimating operating efficiency. Therefore, we argue that the quality of human capital and total team salaries should be included as input variables when estimating the operating efficiency of sports franchises.

Output variables align with the goals of sports franchises, which vary based on the team, sport, and country. Despite these diverse goals, most efficiency studies in the sports domain do not consider both financial and performance aspects [8,28,29,41,47,48]. This research aims to contribute to the efficiency literature in the sports domain by estimating efficiency with more comprehensive input and output variables reflective of sports franchise characteristics. Specifically, we adopt game performance and financial performance as output variables, advancing efficiency studies in the sports domain.

## The role of operating efficiency on franchise value

Based on RBT, we contend that the resources of a sports franchise include the amount of financial investment in the team, the quality of human assets, and operating efficiency. The level of capital invested in a sports team and the human assets of the team are clear core competencies of sports teams. Higher pay levels are one of the strongest determinants of attracting valuable human assets, and preeminent players can contribute to the success of sports teams [49–52]. Furthermore, star players' better game performance can lead to better win probability, attract more fans and media, and sponsors, which can lead to better financial returns [8,9,53–55].

Whether operating efficiency is another core competency of sports franchises, in addition to total salaries and human assets, is a question that remains to be answered. Despite operating efficiency being estimated through variables representing capital, human assets, game, and financial performance, we posit that it is conceptually and empirically distinct from other core competencies. This distinction arises from our approach, where we move beyond a simple calculation of ratios and consider the team's ability to transfer resources into performance, aligning with previous operating efficiency studies [24].

Not only conceptually, the source of operating efficiency is also distinct from other core competencies. The sources of capital and human assets are relatively clear, as they are tangible resources and generally determined by the organization's decision-makers [38,56–58]. However, the source of operating efficiency is more ambiguous. It can come from the system of organization, mentality of the members, organizational culture that is developed over time or transferred by coaching staffs, or even outside of the organizations. In this sense, operating efficiency is an organization's unique resource that is valuable, rare, inimitable, and non-substitutable and even the members of the teams do not know where it is from [24,33,59–61]. Therefore, there should be no misunderstanding that operating efficiency and other core competencies are overlapping concepts simply because capital and human assets are used to estimate operating efficiency. This is because operating efficiency is a more complex concept that goes beyond the simple calculation of ratios. It measures the organization's ability to use its resources effectively and efficiently to achieve its goals.

Aligned with the core premise of RBT, organizational resources contribute to better organizational performance. Operating efficiency, therefore, plays a critical role in the key performance of sports franchises, as evidenced by the use of team efficiency in explaining the

performance of professional sports teams in various studies [24,27,28,62–64]. However, to establish whether operating efficiency constitutes a distinguishable and critical core competence for sports franchises, a statistical test is imperative. If operating efficiency is a core competence and can be distinguished from other core competencies, its effects on organizational performance should remain statistically significant even after controlling for other core competencies.

Determining the most comprehensive measure of organizational performance for sports franchises is essential for a rigorous assessment. As the goals of sports franchises extend beyond game performance or financial return, a holistic measure reflecting both dimensions is necessary. We propose that franchise value serves as an excellent indicator of a sports franchise's comprehensive goals, encompassing both game performance and financial return [13,14]. Therefore, we define franchise value as the performance of a sports organization.

In light of this, it is prudent to explore the relationship between operating efficiency and franchise value. However, it is crucial to acknowledge the temporal dynamics inherent in the evaluation. The efficiency of sports franchises in this study will be assessed over a specific period, with the unit of analysis being a team in a given year. In contrast, franchise value accumulates over time, with a specific year's franchise value reflecting the culmination of organizational performance over the years. Consequently, while the operating efficiency of one year may not directly impact the total franchise value, it may exert influence on the annual change in franchise value.

Building on this rationale, we hypothesize the following.

**Hypothesis:** There is a positive relationship between the operating efficiency of a sports franchise and the incremental value of the franchise, controlling for the level of capital invested and the quality of human assets

## Methodology

In this study, NBA teams serve as sample of sports organizations due to their prominence in the sports industry, offering substantial on- and off-field data. NBA teams operate under various league regulations, including the draft and salary cap [49,65,66]. The input and output factors for NBA teams, used to estimate operating efficiency, exhibit minimal variation, making Constant Returns to Scale (CRS) an appropriate measure for assessing teams' operating efficiency.

The evaluation of organizations' operating efficiency in sports commonly employs data envelopment analysis (DEA), an excellent method for estimating as it does not require statistical assumptions. It compares each decision-making unit (DMU)'s efficiency by comparing the input-to-output ratio among the DMUs. If a DMU has the highest efficiency among the DMUs, the DMU's efficiency is 1. If a DMU's efficiency is extremely low, the efficiency is 0. In this case, a DMU refers to each sports organization. Two types of DEA are prominent: Constant Returns to Scale (CRS) and Variable Returns to Scale (VRS). Given the relatively consistent resources invested in and outputs of sports teams, CRS is typically utilized to assess their efficiency [27,65,66].

Efficiency, as determined by DEA, is influenced by factors such as aggregated player skill levels, total pay-to-game ratio, and financial performance indicators. It is evident that operating efficiency, as estimated through DEA, can impact both game performance and financial outcomes. However, sports organizations may have diverse objectives. Some teams prioritize winning championships, while others, often sponsored by business firms, focus on leveraging sports organizations as marketing tools, placing less emphasis on financial returns. Conversely, certain organizations prioritize financial gains, seeking to boost ticket sales and advertising revenue. Most sports organizations likely fall somewhere between these extremes [50,67,68].

**Table 1. Variables employed for data envelopment analysis.**

| Perspective | Input variables | Output variables |
|---|---|---|
| **Game related variables** | Drive and Pass Percentage, Secondary Assists, Potential Assists, Passes Made, Points by Catch and Shoot, Points by Screen Assists, Deflections, Loose Balls Recovered, Contested Shots | Winning percentage, Final Appearance, Championship Won |
| **Financial related variables** | Total Compensation | Total Revenue, Ticket Revenue, Operating Income |

Data for this study were sourced from various websites that provide information freely accessible to the public. Specifically, game-related data for 30 NBA teams (n = 180) over six seasons (2015~2016 to the 2020~2021 season) were collected from the NBA official website and Basketball-Reference.com. Financial data, including total revenue, ticket revenue, and operating income, were gathered from runrepeat.com, and total compensation was collected from hoopshype.com. Franchise values were obtained from forbes.com. We employed R software to conduct DEA and statistical computing such as regression analysis. The study adopts CRS efficiency as the measure of operating efficiency for NBA teams. Table 1 outlines the variables used in DEA, while Table 2 provides a more detailed overview of these variables.

The study also takes advantage of tracking stats instead of traditional stats for a comprehensive overview of the collective quality of NBA players Tracking statistics go beyond individual player capabilities, representing the combined skills that coaches aim to showcase during games. These statistics encompass drive and pass percentages, secondary assists, potential assists, passes made, points from catch-and-shoot plays, points from screen assists, deflections, loose balls recovered, and contested shots. The financial investment in NBA teams is operationalized as total compensation, while game-related output variables for operating efficiency include winning percentage, final appearances, and winning championships. Financial-related

**Table 2. Definitions of variables.**

| Variables | Variable Definition | Definition Source |
|---|---|---|
| Drive and pass percentage | Pass percentage after driving to the basket. | N/A |
| Secondary assists | "A player is awarded a secondary assist if they passed the ball to a player who recorded an assist within 1 second and without dribbling" | NBA.com |
| Potential assists | "The number of total passes made by a player or team per game The number of total passes made by a player or team per game" | |
| Points by catch and shoot | "The number of points scored by a player or team on Catch and Shoot shots" | |
| Points by screen assists | "The number of times an offensive player sets a screen for a teammate that directly leads to a made field goal by that teammate" | |
| Deflections | "Any time a defensive player gets his hand on a ball in a non-shot attempt" | |
| Loose balls recovered | "The number of times a player or team gains sole possession of a live ball that is not in the control of either team" | |
| Contested Shots | "The number of times a defensive player or team closes out and raises a hand to contest a shot prior to its release" | |
| Winning percentage | Winning percentage of a team after a single season | N/A |
| Final Appearance | Whether a team has reached the finals in a corresponding season | |
| Total compensation | The total amount of spending on players' payroll by a team in a corresponding season | |
| Winning the championship | Whether a team has won a championship in a corresponding season | |
| Total revenue | The total revenue made by a team in a corresponding season | |
| Ticket revenue | The total ticket sales revenue made by a team in a corresponding season | |
| Operating income | A team's profit after managing the season | |

output variables comprise total revenue, ticket revenue, and operating income. Subsequent to the Constant Returns to Scale (CRS) estimation, a regression analysis was conducted to elucidate the impact of operating efficiency on organizational value.

The Forbes organizational brand value is operationalized as the measure of NBA teams' organizational values in this study. Specifically, we utilize the incremental value of organizations' brands as the dependent variable, recognizing that a single-year resource allocation does not singularly determine total organizational value but contributes to incremental value. In this equation, the input variables employed in DEA serve as control variables.

Following CRS estimation, a regression analysis will be employed to test the hypotheses. If CRS demonstrates a significant effect in increasing the marginal value of a sports franchise, while controlling for other variables, we can infer that operating efficiency constitutes a core competency for sports franchises, distinguishable from other competencies.

## Results

Descriptive statistics and intercorrelations among the focal variables are described in Table 3. Focal variables generally show the correlations with what we expected. Specifically, incremental value is strongly related to operational efficiency (r = .39, p < .001), total revenue (r = .68, p < .001), ticket revenue (r = .56, p < .001), final appearance (r = .26, p < .001), championship won (r = .28, p < .001), and operating income (r = .56, p < .001). Operational efficiency also has positive relationships with winning percentage (r = .55, p < .001), total revenue (r = .51, p < .001), ticket revenue (r = .60, p < .001), final appearance (r = .38, p < .001), championship won (r = .27, p < .001), and operating incomes (r = .44, p < .001).

Fig 1 shows the positions of the teams analyzed in this study based on winning percentage, total revenue, and operating efficiency. The size of the circles indicates the level of operating efficiency, with larger circles representing higher operating efficiency. The teams with perfect operating efficiency tend to have better winning percentages and total revenues, but this is not always the case. For example, the Toronto Raptors in 2016 had a relatively good game performance but their financial return was not impressive. The Los Angeles Lakers in 2016 were the opposite. Their total revenue was relatively good among the sample, but their winning percentage was almost disastrous. This implies that operating efficiency is fundamentally related to game performance and financial return, but it is a different concept.

Among the 180 DMUs, 29 DMUs' CRS were 1. This means 29 out of 180 NBA teams had perfect operational efficiency compared to the samples. The list of the NBA teams is described on Table 4.

To explore the further meaning of operating efficiency, we conducted a regression analysis to test our hypothesis.

Prior to conducting the regression analysis, Variable Inflation Factor (VIF) tests were performed, leading to the exclusion of three variables—contested shots, loose balls recovered, and deflections—due to their elevated VIF scores (33.37, 19.19, and 11.859, respectively). The remaining variables exhibited acceptable VIF levels ranging from 1.22 to 4.61. The dependent variable in the regression analysis is the annual increase in team value, as determined by Forbes' annual franchise valuations. The results of the regression analysis are presented in Table 5. In Model 1, various control variables, such as human assets, the budget of the year, and the population increase of the year, were included. This was undertaken because the dependent variable represents the annual increase in the team's value, and changes in city conditions, such as population change, may influence the franchise value change in the city. Model 2 elucidates the role of operating efficiency in predicting incremental organizational value (B = 462.21, p < .001), even after controlling for human assets and budget. Moreover, the

**Table 3. Descriptive statistics and intercorrelation.**

| Variable | M | SD | 1 | 2 | 3 | 4 | 5 | 6 | 7 | 8 | 9 | 10 | 11 | 12 | 13 | 14 | 15 | 16 | 17 |
|---|---|---|---|---|---|---|---|---|---|---|---|---|---|---|---|---|---|---|---|
| 1.Incremental Value | 180.60 | 153.63 | - | | | | | | | | | | | | | | | | |
| 2. Operational Efficiency | 0.79 | 0.15 | .39*** | - | | | | | | | | | | | | | | | |
| 3. Winning Percentage | 0.49 | 0.15 | .13 | .55*** | - | | | | | | | | | | | | | | |
| 4. Total Revenue | 246.86 | 65.29 | .68*** | .51*** | .03 | - | | | | | | | | | | | | | |
| 5. Total Compensation | 109450217.26 | 21260379.53 | .13 | -.14 | .23** | .27*** | - | | | | | | | | | | | | |
| 6. Ticket Revenue | 40.43 | 26.69 | .56*** | .60*** | .02 | .78*** | -.13 | - | | | | | | | | | | | |
| 7. Final Appearance | 0.07 | 0.26 | .26*** | .38*** | .40*** | .28*** | .22** | .17* | - | | | | | | | | | | |
| 8. Championship Won | 0.04 | 0.19 | .28*** | .27*** | .29*** | .23** | .13 | .15 | .69*** | - | | | | | | | | | |
| 9. Operating Income | 50.19 | 40.92 | .56*** | .44*** | -.08 | .79*** | -.10 | .71*** | .02 | .01 | - | | | | | | | | |
| 10. Drive & Pass % | 35.20 | 4.59 | -.18*** | -.26*** | .08 | -.08 | .38*** | -.32*** | -.01 | .01 | -.09 | - | | | | | | | |
| 11. Secondary Assists | 2.94 | 0.54 | .19* | .00 | .40*** | .10 | .32*** | -.08 | .36*** | .29*** | -.03 | .25*** | - | | | | | | |
| 12. Potential Assists | 45.60 | 3.13 | .08 | -.13 | .17* | -.06 | -.04 | -.07 | .29*** | .18* | -.07 | .19* | .51*** | - | | | | | |
| 13. Passes Made | 293.66 | 19.76 | .26*** | -.04 | -.11 | .10 | -.28*** | .17* | .02 | .02 | .18* | -.07 | .37*** | .33*** | - | | | | |
| 14. Catch and Shoot Points | 28.44 | 3.10 | .10 | -.03 | .40*** | .09 | .35*** | -.13 | .39*** | .22** | -.05 | .34*** | .56*** | .60*** | .12 | - | | | |
| 15. Screen Assists points | 18.27 | 8.68 | .13 | -.24** | .02 | .30*** | .66*** | -.06 | .05 | .01 | .13 | .17* | .13 | -.11 | -.16* | .22** | - | | |
| 16. Deflections | 12.03 | 5.52 | .14 | -.29*** | .00 | .29*** | .62*** | -.04 | .02 | .05 | .16* | .24** | .14 | -.03 | -.15* | .25** | .90*** | - | |
| 17. Loose Balls Recovered | 6.22 | 2.91 | .27*** | -.20* | -.02 | .42*** | .60*** | .14 | .01 | .02 | .27*** | .18* | .10 | -.10 | -.10 | .18* | .85*** | .92*** | - |
| 18. Contested Shots | 50.70 | 23.05 | .19* | -.22** | -.02 | .38*** | .59*** | .08 | .02 | .03 | .23** | .19* | .12 | -.12 | -.12 | .20* | .91*** | .95*** | .97*** |

*p < .05

** p < .01

***p < .001.

model fit was significantly improved (delta R-square = 0.15, p < .001; delta BIC = 34.29, p < .001). The results underscore that NBA teams' operating efficiency embodies unique features not mirrored by other organizational resources, offering an incremental explanation for changes in organizational value, emphasizing the distinctive contribution of operating efficiency in influencing the annual increase in team value.

Fig 2 visually represents the intricate relationships among revenue, operating efficiency, and team value increment. Red collars signify the highest value increments, while blue collars denote the lowest value increments. Notably, a discernible pattern emerges wherein a larger area of red collars corresponds to higher CRS efficiency, and conversely, a larger area of blue collars aligns with lower CRS efficiency. This visually compelling evidence solidifies the clear relationship between operating efficiency and team value increment.

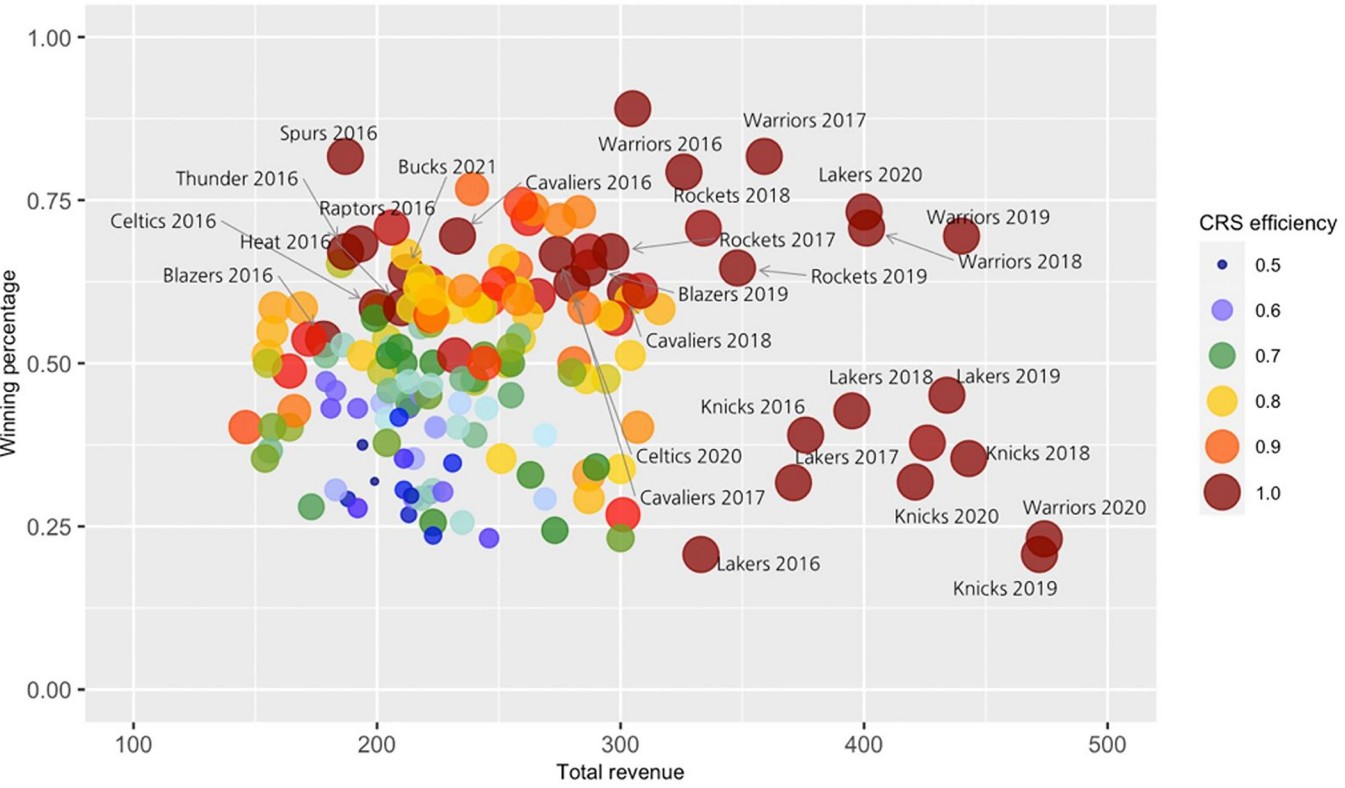

**Fig 1. The relationship among winning percentage, total revenue, and CRS efficiency.**

## Discussion and conclusion

This study provides an empirical exploration of the operating efficiency of professional sports teams, considering both game-related and financial factors. The Data Envelopment Analysis (DEA) results, based on data from 30 NBA teams over six seasons (n = 180), reveal that only 29 out of 180 teams (16%) attained perfect operating efficiency within the sample. Notably, this finding underscores the significant observation that, within the NBA's postseason structure featuring 16 competing teams annually, only three teams can theoretically achieve perfect efficiency each season. This arithmetical constraint emphasizes the formidable challenge of attaining 100% efficiency when considering both game performance and financial outcomes. Additional detailed information on the operationally efficient teams in the NBA league can be found in Table 6.

Subsequently, the study delves into the analysis of the role of operating efficiency in influencing organizational value—a more comprehensive variable that encompasses both game

**Table 4. Efficiency 1 teams in each season.**

| Season | List of teams with efficiency 1 |
|---|---|
| 2020~2021 | Bucks |
| 2019~2020 | Celtics, Knicks, Lakers, Warriors |
| 2018~2019 | Blazers, Knicks, Lakers, Rocket, Warriors |
| 2017~2018 | Cavaliers, Knicks, Lakers, Rockets, Warriors |
| 2016~2017 | Cavaliers, Lakers, Rockets, Warriors |
| 2015~2016 | Blazers, Cavaliers, Celtics, Heat, Knicks, Lakers, Raptors, Spurs, Thunder, Warriors |

**Table 5. The result of regression analysis.**

| | Model 1 | | | | Model 2 | | | |
|---|---|---|---|---|---|---|---|---|
| | **B** | **SD** | **t** | **p** | **B** | **SD** | **t** | **p** |
| **Constant** | -298.29 | 244.74 | -1.22 | 0.22 | -1132.55 | 255.29 | -4.44 | 0.00 |
| **Drive&pass %** | -10.53 | 2.56 | -4.12 | 0.00 | -5.94 | 2.40 | -2.47 | 0.01 |
| **Secondary assists** | 31.23 | 27.56 | 1.13 | 0.26 | 3.60 | 25.14 | 0.14 | 0.89 |
| **Potential assists** | 1.54 | 4.66 | 0.331 | 0.74 | 8.52 | 4.32 | 1.97 | 0.05 |
| **Passes made** | 1.38 | 0.64 | 2.16 | 0.03 | 1.9 | 0.58 | 3.28 | 0.00 |
| **Catch and shoot points** | 3.59 | 4.98 | 0.72 | 0.47 | -1.94 | 4.56 | -0.43 | 0.67 |
| **Screen assists points** | -7.95 | 2.50 | -3.18 | 0.00 | -5.07 | 2.29 | -2.21 | 0.03 |
| **Loose balls recovered** | 30.55 | 6.98 | 4.38 | 0.00 | 28.04 | 6.28 | 4.47 | 0.00 |
| **Compensation** | 0.00 | 0.00 | 1.51 | 0.13 | 0.00 | 0.00 | 1.98 | 0.05 |
| **Population Increase** | -0.00 | 0.00 | -0.59 | 0.55 | 0.00 | 0.00 | 0.12 | 0.90 |
| **Operating efficiency** | | | | | 462.21 | 71.79 | 6.44 | 0.00 |
| **Model p value** | 0.00 | | | | 0.00 | | | |
| **Adjusted R-squared** | 0.21 | | | | 0.36 | | | |
| **Delta R-squared** | | | | | 0.15 *** | | | |
| **BIC** | 2330.68 | | | | 2296.39 | | | |
| **Delta BIC** | | | | | 34.29 ** (> 10) | | | |

***$p < .001$.

and financial performance, aligning with the overarching goals of most sports organizations. The results underscore that the operating efficiency of NBA teams significantly impacts their value. This implies a critical strategic implication for sports organizations, suggesting that a focus on operating efficiency is paramount for achieving their fundamental goals.

This study holds several theoretical implications. First, it introduces a comprehensive approach by incorporating both game and financial aspects of sports organizations conducting data envelopment analysis. Conceptualizing professional sports organizations as not only

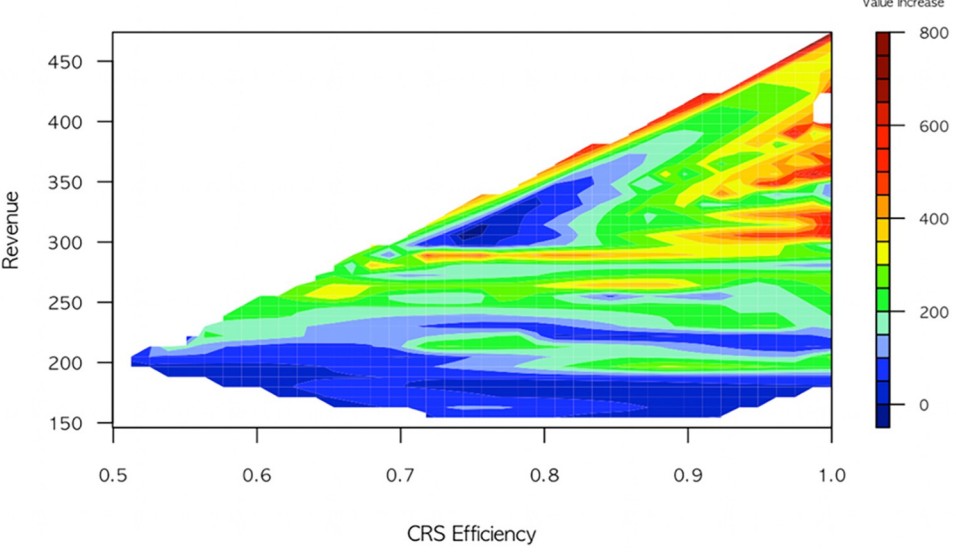

**Fig 2. The relationships among revenue, operating efficiency, and Team value increased.**

**Table 6. The detailed information of NBA teams with perfect efficiency.**

| Season | Teams | Data & Narrative analysis | Financial note from Forbes |
|---|---|---|---|
| 2016 | Blazers | · Not only Lamarcus Aldridge left the Blazers, Nicolas Batum, Robin Lopez, and Wesley Matthews who were in the starting line-ups moved on to the new team, but this season was when C.J McCollum won the MIP award and proved he was ready to partner up with team's star Damian Lillard.<br>· Despite ESPN's projection of 31 wins for the Blazers, the 2016 team, with the lowest payroll in our dataset, achieved a winning percentage that ranked 67th out of 168 teams, securing 44 wins. | "Season ticket renewals were 97% and ranked fifth in the NBA this season. New additions to the Moda Center this year include a refurbished courtside lounge and a kid-friendly activity area. The Trail Blazers are in line for a new TV deal with their pact with Comcast SportsNet Northwest set to expire after the 2016–17 season." |
| 2016 | Cavaliers | · The front office had to dismiss Coach David Blatt due to his inability to maintain cohesive chemistry in the locker room. As players supported their assistant coach Tyron Lue over Blatt, Lue was replaced as head coach in the mid-season. The Cavaliers then secured a historic comeback from a 3–1 deficit to win the championship. | "LeBron James' return to the Cavaliers in 2014 was a boon for the club's bean counters. The Cavs didn't raise ticket prices for the 14–15 season, but arena revenue still surged more than 40% thanks to increased attendance and sponsorships, as well as a trip to the NBA Finals. Attendance jumped 17% to second in the league behind the Bulls" |
| 2016 | Celtics | · The Celtics, despite having the 16th lowest payroll, concluded the season with a commendable 59% winning percentage, ranking 44th in the dataset.<br>· The 2016 season marked Boston's first without Paul Pierce, Kevin Garnett, and Rajon Rondo. Nevertheless, the Celtics emerged as a winning team, propelled by Isaiah Thomas's outstanding performance. | "The Celtics are still among the most profitable teams in the NBA. The team pays no rent at TD Garden and pulls in over $60 million a year in local television revenue, including its equity stake in Comcast SportsNet New England." |
| 2016 | Heat | · Despite the Heat having the 28th lowest payroll, they achieved a 59% winning percentage, ranking 44th in the dataset.<br>· After missing the playoffs in their first year without LeBron James, the Heat rebounded and reached Game 7 in the Eastern Conference semifinals. | "The Heat's first year without LeBron James was a drastic change as the team failed to make the playoffs after four straight appearances in the NBA Finals. Fans still supported the team with ratings on TV down only 3% and third highest in the league. Arena attendance also ranked sixth overall." |
| 2016 | Knicks | · Phil Jackson attempted to implement the Triangle offense for the Knicks, which proved unsuccessful, ultimately leading to the mid-season firing of Coach Derek Fisher.<br>· Despite the 2015–16 season being one of the worst in the Carmelo Anthony era in New York, the Knicks ranked 3rd, 8th, and 11th in ticket revenue, operating income, and total revenue in our dataset. | "The split of the media and sports assets of Madison Square Garden Company in September precipitated a new media rights deal for the Knicks with the MSG regional sports network. The 20-year pact kicks off this season and is worth $100 million in the first year." |
| 2016 | Lakers | · The Lakers finished last in the Western Conference but still managed to secure 8th, 11th, and 16th positions in ticket revenue, operating income, and total revenue in our dataset.<br>· In the 2015–16 season, Byron Scott faced challenges in his relationship with players. D'Angelo Russell criticized Scott for mismanagement after leaving the Lakers. Tracking stats also highlight the Lakers' deficiencies, ranking lowest in secondary assists, 147th in passes made, and 167th in both potential assists and passes made. | "The Lakers are the NBA's most profitable team thanks to the team's 20-year, $3.6 billion deal with Time Warner Cable's SportsNet LA. Ratings were off more than 50% for Lakers' games during the 2014–15 season, but the average audience size of 122,000 viewers per game was still the second highest in the league." |
| 2016 | Raptors | · The 2016 Raptors achieved the 17th highest winning percentage in the dataset with the 163rd payroll.<br>· Despite finishing 2nd in the East, their rankings of 164th in potential assists, 159th in secondary assists, and 117th in passes from the drive indicate potential shortcomings in Coach Dwane Casey's offensive scheme. | "The Raptors sold out season tickets last year for the first time since the 2000–01 season. Season ticket sales jumped 50% to 12,500. The team also added new sponsors Sun Life Insurance, Sentry Investments, Unilever and Aeolus Tires." |
| 2016 | Spurs | · LaMarcus Aldridge joined the Spurs in 2016, finishing the season with an 82% winning percentage, ranking 2nd in the West and 2nd highest in our dataset.<br>· They excelled in passes made (7th) and secondary assists (10th), showcasing effective ball movement. | "The Spurs led the NBA in local television ratings during the 2014–15 season, posting an average of 8.09 on FS Southwest, 9% higher than the previous year. The Spurs completed a $110 million renovation of AT&T Center for this season that includes new seats, video boards and 14 theater boxes." |
| 2016 | Thunder | · Kevin Durant's last season with the Thunder in 2016 resulted in a 67% winning percentage in our dataset. The Thunder reached the Western Conference finals and nearly eliminated the 73–9 Warriors.<br>· The team ranked 3rd and 10th lowest in secondary assists and passes made, respectively. This could be an explanation of why Kevin Durant was frustrated with his former team, leading him to leave for the Warriors even though the Thunder had a high winning percentage. | "The Thunder once again played in front of a packed-house during the entire 2014–15 season as fans continued to fill Chesapeake Energy Arena. After registering sellouts during all 41 home games in each of the four previous seasons, the Thunder has recorded sellouts in 187 consecutive regular season home outings and 222 games overall, including the postseason." |

*(Continued)*

**Table 6.** (*Continued*)

| Season | Teams | Data & Narrative analysis | Financial note from Forbes |
|---|---|---|---|
| 2016 | Warriors | · The 2016 Warriors achieved the highest winning record in NBA history with 73 wins and 9 losses. The team ranked 6th in secondary assists, 2nd in potential assists, 12th in passes made, and 3rd in points by catch and shoot.<br>· They also ranked 21st and 25th in total and ticket revenue, respectively, with a corresponding 35th position in operating income. Notably, these achievements were attained with a payroll ranking as low as 128th in the dataset. | "In-arena merchandise sales doubled for the Warriors last season and the team sold out every game for the second season in a row. The Warriors have more than 10,000 names on their season ticket waiting list. The Warriors proposed new arena took another step forward in December when the San Francisco Board of Supervisors approved the environmental report for the multi-use arena planned for San Fran's Mission Bay neighborhood." |
| 2017 | Cavaliers | · The 2017 Cleveland team ranked 40th in total revenue and 30th in winning percentage. Despite finishing 2nd in the east and reaching the finals.<br>· This season marked the departure of Kyrie Irving, possibly indicating a chemistry issue in the locker room. However, even without considering the chemistry issue, the 2017 Golden State Warriors were formidable in the finals, particularly with the addition of Kevin Durant in the summer. | "The Cavaliers captured the team's first NBA title in 2016 with a dramatic seven-game series comeback against the defending champion Golden State Warriors. The Cavs had the NBA's second highest average local cable ratings on Fox Sports Ohio. In addition, the Cavaliers posted the second-highest attendance in the league for the second straight year. Despite the season-long sellout and playoff run, the Cavs had the fifth biggest loss in NBA history due to their massive payroll." |
| 2017 | Lakers | · The incident of D'Angelo Russell exposing a private conversation with Nick Young had caused a locker room issue during the season. The Lakers ended up trading Russell next summer and did not resign Young.<br>· The Lakers finished the season with 14th in the west. However, they were 12th in total revenue, 7th in ticket revenue, and 9th in operating income with relatively low payroll which is 42nd in our dataset. | "Last season marked the end of an era with the retirement of Kobe Bryant, who spent 20 years in purple and gold and finished third all time in NBA scoring. It was the worst season in franchise history on the court with a 17–65 record (the previous year was second worst), but the team still cashed in with fans eager to get one last glimpse of No. 24." |
| 2017 | Rockets | · The Rockets ranked 29th in total revenue and 24th in operating income, with 36th lowest payroll<br>· The Rockets were 3rd in the west with the 19th highest winning percentage. With Mike D'Antoni's arrival as a head coach, the Rockets brought a faster pace to the game from 7th to 3rd and averaged 2nd most points per game in the league as well as 2nd in offensive rating. Within D'Antoni's system, James Harden finished 2nd in points and 1st in assists and finished 2nd in MVP voting. | "The Rockets will have one of the league's best players under wraps for several years after signing James Harden to a contract extension in July that runs through the 2019–20 season and is worth $118 million. Harden, who had two years remaining on his original deal with the Rockets, will receive an additional $20 million for those two seasons plus $63 million for the two additional years. Harden is the leading MVP candidate approaching the All-Star break." |
| 2017 | Warriors | · The 2017 Warriors is known as one of the best teams in NBA history. Our data also shows how well they played the game systematically. The Warriors were 2nd in secondary assists, 1st in potential assists, 1st in points by catch and shoot, 2nd in points by screen assists, and 5th in contested shots.<br>· The Warriors were 13th in total revenue, 3rd in winning percentage, and won a championship against the defending champion. | "On its way to the best regular season record in NBA history, the team posted the league's highest average cable TV rating (9.76) during the 2015–16 season, more than double the previous season. The value of the Warriors is up a league-high 37% and the team could challenge the Lakers and Knicks for the NBA's highest revenue in their new arena, which broke ground in January. The season ticket waiting list was at 32,000 to start the current season with the renewal rate at 99.5%." |
| 2018 | Cavaliers | · Kyrie Irving left Cleveland, and the most of the tracking stats fell compared to the year before and finished 4th in the east. The Cavaliers had a rough season blending the new players to the team, including Dwyane Wade, Derrick Rose, Isaiah Thomas, and Jae Crowder and ended up trading all of them in the mid-season. However, Cleveland again went on to the finals with newly acquired role players, including Jordan Clarkson and Larry Nance Jr.<br>· In the finals, LeBron James averaged 34.0 points, 10.0 assists, and 8.5 rebounds, but same as the last year's finals, Kevin Durant and the Warriors were too formidable as an opponent for the Cavs and could not avoid a sweep. | "The Cavaliers continue to be a massive draw thanks to four-time MVP LeBron James. The team had the second-highest home attendance and led the league in road attendance during the 2016–17 season. Local TV ratings averaged 7.38 on Fox Sports Ohio and ranked second-best in the NBA, although they were down 21% from the previous season when the team captured its first NBA title." |
| 2018 | Knicks | · The Knicks 134th in winning percentage, but were 3rd in total revenue, 6th in ticket sales, and 4th in operating income.<br>· This was the Knicks' first season after they traded away Carmelo Anthony and fired Phil Jackson from the front office. Despite the team rebuilding, the fans' attendance in Madison Square Garden was 9th in the league. | "The Knicks are the NBA's most valuable team for the third straight year. The Knicks have been shut out of the playoffs since 2013, but they are reaping the rewards of a $1 billion renovation to Madison Square Garden, which produced new revenue opportunities." |
| 2018 | Lakers | · After trading away D'Angelo Russell, and adding young cores, including Kyle Kuzma, Lonzo Ball, Josh Hart, and Alex Caruso, Lakers' winning percentage slightly increased from 32% to 43%.<br>· The Lakers finished the season with 11th in the west. Nonetheless, the Lakers were 10th in total revenue, 5th in ticket revenue, and 7th in operating income. | "The Lakers remain one of the NBA's glamour franchises and one of its most profitable thanks to nearly $150 million a year from its local TV and radio deals. It is 10 to 15 times what some small market teams bank. The last four seasons have ranked among the five worst in franchise history for winning percentage." |

(*Continued*)

**Table 6.** (*Continued*)

| Season | Teams | Data & Narrative analysis | Financial note from Forbes |
|---|---|---|---|
| 2018 | Rockets | Rockets' winning percentage was 4th in our data with 67 wins and finished 1st in the west.<br>· Chris Paul joined the team, and James Harden ended up winning an MVP award. With the two stars, the Rockets almost eliminated the Warriors in the conference finals, and Houston was the only team that forced Golden State to game 7, while all other series were done in game 5. | "The NBA approved Tilman Fertitta's $2.2 billion bid for the Rockets in October. The price was a record for a sports franchise. Fertitta made his fortune through the Golden Nugget Casinos and Landry's, a Texas based restaurant and entertainment company. He originally tried to buy the team in 1993, but his $81 million bid fell short to the $85 million by Leslie Alexander." |
| 2018 | Warriors | Warriors were able to re-signed with their key bench players including David West, Andre Iguodala, and Shaun Livingston.<br>· The Warriors were 8th in total revenue, 10th in ticket revenue, and 17th in operating income out of 168 teams. At the same time, GSW finished with 71% in winning percentage(13th in our dataset), and repeated as NBA champion. | "The defending NBA champs raised ticket prices from 15% to 25%, depending on location, for the 2017–18 season. Price hikes are easy with an NBA-high 40,000-plus season ticket waiting list. The Warriors' three-year, $60 million jersey patch deal with Rakuten is the richest for any NBA team." |
| 2019 | Blazers | This season was the Blazer's highest winning percentage in our data, and 9th highest in franchise history. Portland made its first conference finals appearance since 2000. However, Damian Lillard had to play through separated ribs without injured Jusuf Nurkic against the Warriors, and fell short to advance to the finals.<br>· While Al Farouq-Aminu and Maurice Harkless were key players in Portland's defensive end, assistant coach David Vanterpool played a significant role in its scheme. Vanterpool later left the team and was hired as an Associate head coach and worked as a defensive coordinator for the Timberwolves. The Blazers' defense plummeting when the two players and Vanterpool left the following season shows that they were the defensive linchpins for the Blazers' success in 2019. | "Portland Trail Blazers kicked off new local TV deals last season at more than double their previous take." |
| 2019 | Knicks | The Knicks tied their worst franchise record with 17 wins. The team's star, Kristaps Porzingis, could not play due to an ACL injury, and to make matters worse, Porzingis requested a trade during the season. The front office might have intended to tank for Duke's star Zion Williamson and aimed to empty the salary so that they could sign two max stars next summer.<br>· Nonetheless, the Knicks were 2nd in all financial categories in our span. | "Owner James Dolan continues to spin straw into gold with the NBA's most valuable team for the fourth-straight year. The Knicks' $1 billion renovation to Madison Square Garden, completed in 2013, produced new revenue opportunities. Despite nearly two decades of losing, the Knicks command premium prices for tickets, suites and sponsorships. The team's local cable deal with MSG, worth more than $100 million a year, is the second richest in the sport behind that of the Los Angeles Lakers." |
| 2019 | Lakers | · The Lakers signed LeBron James and other free agents including Kentavious Caldwell-Pope, Rajon Rondo, and Lance Stephenson in the summer. However, the Lakers had to shut down LeBron due to an injury before the season ended, and eventually finished 10th in the west.<br>· The Lakers did not improve much from game-wise, but were 1st in ticket revenue, 3rd in operating income, and 5th in total revenue in our sample. | NA |
| 2019 | Rockets | · Trevor Ariza's departure was considerable to the Rockets' defense, as there weren't any players who could fill up the absence of the two-way swingman, Rockets' defensive rating dropped from 6th in the league to 17th. James Harden still averaged more than 40 points per game for a 40-game span, and the Rockets were able to be 23rd in ticket revenue and 14th in both total revenue and operating income in our data. | NA |
| 2019 | Warriors | · From stat-wise, the Warriors did not fall compared to the last season, including the offensive and defensive ratings. The Warriors were the favorite to win the championship from the beginning and finished 1st in the west.<br>· Two main issues that hindered the Warriors from a 3-peat were an on-court feud between Kevin Durant and Draymond Green and the injuries that prevented Thompson and Durant from playing in the finals. | "The Golden State Warriors are leaving Oracle Arena, the NBA's oldest building, next season for the $1 billion Chase Center. The reigning league champs have secured $2 billion in contractually obligated income from sponsorships, suites and season ticket holder fees for the new arena." |
| 2020 | Celtics | · Kyrie Irving walked out of Boston as a Free Agent. However, Jaylen Brown stepped up from 13.0 to 20.3 points per game player, and Jayson Tatum from 15.7 to 23.4, which led the Celtics to finish 3rd in the east.<br>· Boston were 4th in defensive rating in the league and 9th in deflections in our 168 data points. | NA |

(*Continued*)

**Table 6.** (Continued)

| Season | Teams | Data & Narrative analysis | Financial note from Forbes |
|---|---|---|---|
| 2020 | Knicks | · The Knicks fired coach Fizdale in just 22 games of his second season regardless of his four-year $22 million contract. The front office was not patient with a 4–18 record, as they had brought multiple free agents, including Julius Randle, Taj Gibson, Bobby Portis, Wayne Ellington, Marcus Morris, and Reggie Bullock.<br>· The Knicks still finished 12th in the east but were 7th in total revenue and 13th in ticket revenue in the 168 samples. | NA |
| 2020 | Lakers | · The Lakers finalized a block-buster trade in the summer by sending three young cores of Brandon Ingram, Lonzo Ball, Josh Hart, and three future first-round draft picks for Anthony Davis.<br>· Anthony Davis effect was everything LeBron expected and more. The Lakers' defensive rating were 3rd in the league with the 11th deflection in our dataset, and won their 17th championship in the bubble. | NA |
| 2020 | Warriors | · The Warriors started the season with D'Angelo Russell as Kevin Durant went to the Nets and Klay Thompson's torn ACL injury. Later in the season, the Warriors traded Russell for Andrew Wiggins. Thus, the Warriors were very unstable during the season in many ways. However, Golden State was still 1st in total revenue and operating income in the 2020 season. | NA |
| 2021 | Bucks | · Many fans and the media thought the Buck gave away too much for Jrue Holiday. Milwaukee sent out 4 draft picks, then signed a 4-year 160 million contract with Holiday, which led the Bucks to have 11th payroll in our sample. However, Jrue's impact on defense was invaluable especially in the playoffs and paid off by Bucks winning their first championship in 47 years. | NA |

athletic groups but also as business units with sports content, this study recognizes the critical importance of both game and financial performance as essential goals. Furthermore, the inclusion of players' aggregated skill level as input variables, in addition to the budget invested, distinguishes this study from most DEA studies in the sports domain [20,69]. By conceptualizing players as human assets, the study contributes to advancing DEA studies in the sports domain, offering a more holistic estimation of operating efficiency.

Second, we adopted tracking stats instead of traditional measures of the NBA for estimating operating efficiency. While traditional measures persist in many studies, this research argues their inadequacy in capturing the collective qualities of human assets within teams. To the best of our knowledge, this is the first study that uses tracking stats to conduct DEA providing a more dynamic and nuanced perspective.

Third, we analyzed how operating efficiency works in sports organizations when various goals are considered. Achieving a better position with outstanding game and financial performance have common but different aspects. However, the value of organizations in sports can include both aspects. If efficiency can increase organizational value even after controlling for other significant factors, the organizations must pursue increasing operating efficiency regardless of their different goals. In this sense, the results of this study show why operating efficiency of sports organizations is important.

This study holds valuable practical implications for sports organizations, emphasizing the imperative to enhance operational efficiency. The argument posits that sports organizations, contingent upon their specific situations, may prioritize diverse goals, ranging from game-related performance to financial success. Notably, organizations sponsored by external businesses with a primary focus on game-related performance operate differently from those pursuing financial objectives. In the former case, the organization seeks to bolster its organizational value for sponsorship purposes. Conversely, organizations prioritizing financial

performance aim to increase revenue through enhanced ticket sales, advertising, and improved organizational value through fan engagement [70]. The call for increased operational efficiency resonates across these varied goals. Even organizations with differing objectives ultimately share the common need to augment their value. Operational efficiency, derived from various sources such as coaching staff, organizational systems, and external factors, plays a crucial role. Noteworthy is the finding that sports organizations facing distinct situations can achieve high levels of operational efficiency through different means. Thus, general managers must discern the specific conditions of their organizations and emulate strategies conducive to enhancing operational efficiency.

The study significantly contributes to the sports research domain by estimating the operating efficiency of sports organizations, considering various factors, and elucidating the pivotal role of operating efficiency in influencing organizational value. However, several limitations warrant consideration. Firstly, given the inclusion of financial aspects in this study, operating efficiency may be susceptible to external factors such as economic cycles. The six-season sample period may lead to operating efficiency estimations influenced by these external factors. Future studies could address this limitation by employing alternative methodologies, such as DEA-Window, which accommodates different time frames [51]. Moreover, advancing this study could involve the consideration of more diverse factors beyond players' collective game performance. In the current analysis, human assets were solely represented by players. However, the impact of coaching staff or front-office personnel on operating efficiency in sports games necessitates future studies to broaden the scope and incorporate other human assets within organizations [71,72]. Lastly, the study acknowledges the potential variation in the effects of operating efficiency on organizational value based on organizational goals. [73] contends that organizational strategy may moderate the relationship between organizational systems and organizational performance. Future research may explore the role of sports organizations' goals as moderators, providing valuable insights into the nuanced dynamics of operating efficiency in sports organizations.

## Author Contributions

**Conceptualization:** Philsoo Kim, Sang Hyun Lee.

**Data curation:** Philsoo Kim.

**Formal analysis:** Philsoo Kim.

**Investigation:** Philsoo Kim, JeongJun Moon.

**Methodology:** Philsoo Kim, JeongJun Moon.

**Project administration:** Philsoo Kim, Sang Hyun Lee.

**Resources:** Philsoo Kim.

**Software:** Philsoo Kim.

**Supervision:** Philsoo Kim.

**Validation:** Philsoo Kim.

**Visualization:** JeongJun Moon.

**Writing – original draft:** Philsoo Kim, Sang Hyun Lee.

**Writing – review & editing:** JeongJun Moon.

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
