## [Decision Letter · Decision Letter 0]

17 Nov 2023

PONE-D-23-32235Evaluating the Operational Efficiency of NBA Teams on Franchise Value: An Assessment of Data Envelopment AnalysisPLOS ONE

Dear Dr. Lee,

Thank you for submitting your manuscript to PLOS ONE. After careful consideration, we feel that it has merit but does not fully meet PLOS ONE’s publication criteria as it currently stands. Therefore, we invite you to submit a revised version of the manuscript that addresses the points raised during the review process.

We look forward to receiving your revised manuscript.

Kind regards,

Haroldo V. Ribeiro

Academic Editor

PLOS ONE

Journal Requirements:

3. We note that you have referenced (Kim, P. S.) which has currently not yet been accepted for publication. Please remove this from your References and amend this to state in the body of your manuscript: (Kim, P. S. [Unpublished]) as detailed online in our guide for authors

4. We notice that your supplementary figures are uploaded with the file type 'Figure'. Please amend the file type to 'Supporting Information'. Please ensure that each Supporting Information file has a legend listed in the manuscript after the references list.

Reviewers' comments:

Reviewer's Responses to Questions

**Comments to the Author**

1. Is the manuscript technically sound, and do the data support the conclusions?

Reviewer #1: Yes

Reviewer #2: Yes

2. Has the statistical analysis been performed appropriately and rigorously? 

Reviewer #1: Yes

Reviewer #2: Yes

3. Have the authors made all data underlying the findings in their manuscript fully available?

Reviewer #1: No

Reviewer #2: Yes

4. Is the manuscript presented in an intelligible fashion and written in standard English?

Reviewer #1: Yes

Reviewer #2: Yes

5. Review Comments to the Author

Reviewer #1: The topic of this manuscript is quite exciting and will be of interest to a general audience.

I have some minor comments:

1. Check the Abstract. The authors should be careful when writing the manuscript and formatting the Heading.

2. The authors mention 180 NBA teams. Are you doing a duplicate count? The NBA has 30 teams.

3. Following up on the above point, even if you consider each team per season as unique, 29 out of 180 is not significant at all. The authors should articulate their work clearly.

4. The Introduction section is lengthy and often repetitive. You should be clear in your objective.

5. How effective is the measure of winning percentage? Winning two matches in 2 games is not the same as winning ten matches in 10 games, where both will have 100% wins.

6. What is your dependent variable? The authors need to explain clearly. Is it an incremental value? If so, how are you measuring it? For the benefit of the readers, you should add a table explaining what each variable means.

7. In Table 4 report the VIF values as well. Also, the increase in Adjusted-R2 seems to be much higher – too much higher values can occur due to multicollinearity as well. So, you should estimate the BIC values and the drop in BIC (See AE Raftery 1995 work https://www.jstor.org/stable/271063)

Reviewer #2: Report on "Evaluating the Operational Efficiency of NBA Teams on Franchise Value: An Assessment of Data Envelopment Analysis."

General comments

I have studied the article entitled “Evaluating the Operational Efficiency of NBA Teams on Franchise Value: An Assessment of Data Envelopment Analysis” submitted for publication in PLOS ONE.

In this work, the authors investigate the hypothesis that yearly changes in franchise value, controlling for the amount of capital invested and the quality of human assets (players), are positively associated with a team’s operational efficiency. To do so, they estimate the operational efficiency of 180 NBA teams (all 30 franchise teams between the 2015-2016 and 2020-2021 seasons), controlling for team-aggregated performance indicators and total compensation. The authors find that operational efficiency is indeed positively associated with changes in franchise value. I only wish the authors to address some minor issues before recommending it for publication.

General issues

While reading this paper, I could not stop thinking about some other variables that might be directly associated with franchise value and that are not mentioned by the authors such as location (coastal cities, large metropolises, small cities) and possibly the total number of titles in a franchise’s history. In this direction, I would like to read from the authors something that justifies the exclusion of such variables in the present analysis or if such variables can be thought to be already encompassed by correlated indicators already present in their work.

While discussing the importance of star players in a team, for instance, the authors do not mention any intrinsic geographical or historical characteristics of cities (franchises) that might help to attract star players or that may drive them away.

Some odd sentences:

- Lines 52-53: “The representation of sports franchises and type of business units are obliged to be efficient seeking not only the goal of profit maximization but also to increase its franchise value” - I did not understand the sentence;

- Lines 258-259: “We collected 180 NBA teams of game related and financial related data from 6 seasons (2015~2016 – 2020~2021 season).” -> We collected game and financial related data from 6 seasons (2015~2016 – 2020~2021 season) and comprising 180 NBA teams???

Spelling issues:

- Line 61: “player-performance” (extra space);

- Line 67: “teams have always been an important agenda” -> have always been?;

- Line 68: “Previous literature” -> I believe “literature” does not have a plural form; it is an uncountable noun;

- Line 82: “human capital” -> I believe “capital” is also uncountable in this context;

- Line 240: "The input and output factors of team in NBA" -> "The input and output factors of teams in the NBA”;

6. PLOS authors have the option to publish the peer review history of their article (what does this mean?). If published, this will include your full peer review and any attached files.

Reviewer #1: No

Reviewer #2: No

---

## [Author Response · Author response to Decision Letter 0]

5 Jan 2024

Revier 1. 

Order Comments Response 

1 Check the Abstract. The authors should be careful when writing the manuscript and formatting the Heading.

 I checked and corrected some mistakes.

2 The authors mention 180 NBA teams. Are you doing a duplicate count? The NBA has 30 teams.

 I understand that the writing was confusing. I revised the sentences to convey the idea of 30 NBA teams over 6 seasons (n=30, or 30 decision-making units).

3 Following up on the above point, even if you consider each team per season as unique, 29 out of 180 is not significant at all. The authors should articulate their work clearly.

 Out of 180 teams, 29 (16%) may appear insignificant at first glance. However, it is essential to consider the unique features of professional leagues. In the NBA, 16 out of 30 teams compete in the postseason, and participating in the postseason holds special significance for professional sports players and teams. Despite this, many teams that played in the postseason were not included in the lists. Notably, even teams that reached the semi-finals did not make the lists. In this study, teams exhibiting perfect efficiency are fewer than three in each season, demonstrating both good game and financial performance efficiently.

To elaborate on this point, I have incorporated the following sentences into the discussion: “The DEA results with 30 NBA teams from 6 seasons (n=180) show that only 29 out of 180 (16%) teams achieved perfect operating efficiency among the sample teams. When considering the NBA's situation where 16 teams compete in the postseason each season, it implies a significant observation that, arithmetically, only three teams can demonstrate perfect efficiency each season. In other words, taking into account both game performance and financial outcomes, achieving 100% efficiency is highly challenging.”

4 The Introduction section is lengthy and often repetitive. You should be clear in your objective.

 I made it much shorter than before by removing abundant sentences 

5 How effective is the measure of winning percentage? Winning two matches in 2 games is not the same as winning ten matches in 10 games, where both will have 100% wins.

 I believe there was a misunderstanding at this point. All teams play an equal number of 82 games in a standard season. The winning percentage is a variable measured after the season ends. We contend that a single season in the NBA provides sufficient time to assess the overall capability of a team. Consequently, there is no difference in the meaning of winning percentage based on the stages of each season.

6 What is your dependent variable? The authors need to explain clearly. Is it an incremental value? If so, how are you measuring it? For the benefit of the readers, you should add a table explaining what each variable means. The dependent variable is an incremental value per year based on Forbes' annual franchise valuations. Forbes calculates these values considering various aspects such as team performance, financial earnings, etc. I have attempted to convey this information in multiple ways throughout the manuscript. Additionally, we have included explanations of the variables used to perform Data Envelopment Analysis (DEA) in Table 2. 

7 In Table 4 report the VIF values as well. Also, the increase in Adjusted-R2 seems to be much higher – too much higher values can occur due to multicollinearity as well. So, you should estimate the BIC values and the drop in BIC (See AE Raftery 1995 work https://www.jstor.org/stable/271063) The VIFs of Contested Shots (33.37), Loose Balls Recovered (19.19), and Deflections (11.89) were excessively high, even though they were considered control variables. Consequently, we sequentially removed each of these three variables until all the VIFs fell below 5. The table presents the results of the VIF test with the remaining variables

Variables Drive & pass % Secondary assists Potential assists Passes made Catch & shoot points Screen assists points Loose balls recovered Compensation Population Increase Operating Efficiency

VIF 1.40 2.14 2.21 1.62 2.44 4.61 3.92 3.24 1.22 1.28

To explain this, we put these sentences: 

“We conducted VIF tests prior to the regression analysis, and three variables (contested shots, loose balls recovered, and deflections) were excluded due to their high VIF scores (33.37, 19.19, and 11.859, respectively). Other variables showed an acceptable VIF level ranging from 1.22 to 4.61” 

We re-conducted the regression analysis with revised control variables. Table 4 presents the results, and the corresponding explanation is as follows:

“The dependent variable in the regression is the annual increase in team value, as determined by Forbes' annual franchise valuations. Table 4 presents the results. In Model 1, we included various control variables such as human assets, the budget of the year, and the population increase of the year. This was done because the dependent variable is the increased value of the team each year. Model 2 shows the role of operating efficiency in predicting incremental organizational value (B=462.21, p<.001) after controlling for human assets and budget. Furthermore, the model fit was significantly increased (delta R-square=0.15, p<.001; delta BIC=34.29, p<.001).”

Reviewer 2

order Comments Response 

1 While reading this paper, I could not stop thinking about some other variables that might be directly associated with franchise value and that are not mentioned by the authors such as location (coastal cities, large metropolises, small cities) and possibly the total number of titles in a franchise’s history. In this direction, I would like to read from the authors something that justifies the exclusion of such variables in the present analysis or if such variables can be thought to be already encompassed by correlated indicators already present in their work.

 Your comments are very insightful, and we carefully considered them, leading us to re-analyze additional data. Subsequently, we have partially incorporated your suggestions.

Firstly, it's crucial to clarify that the dependent variable is not the Forbes franchise value itself but rather the increase in Forbes franchise value. This distinction is important as constant variables, such as location, may impact Forbes franchise value, but they might not directly contribute to the increase in Forbes franchise value.

To illustrate, consider the Golden State Warriors, who ranked 1st in NBA franchise value in 2023. However, their value was 28th out of 29 teams in 2004. Despite the location remaining constant, the franchise value increased from 28th to 1st as the Warriors evolved into one of the greatest teams in NBA history. Conversely, the Detroit Pistons held the 4th position in franchise value but dropped to 23rd in 2023. In the Pistons' case, while the location remained the same, the franchise experienced a shift from a winning culture after 2004 to a rebuilding phase in the 2020s.

We acknowledge that city conditions, especially factors like population size, may influence changes in Forbes franchise values. These changes can reflect fluctuations in potential market size and even the economic situation of the city.

To address this consideration, we have added population changes as a control variable. The following sentences are inserted to reflect this adjustment:

"Table 4 presents the results. In Model 1, we included various control variables such as human assets, the budget of the year, and the population increase of the year. This was done because the dependent variable is the increased value of the team each year. Changes in city conditions, such as population change, may influence the franchise value change in the city."

2 While discussing the importance of star players in a team, for instance, the authors do not mention any intrinsic geographical or historical characteristics of cities (franchises) that might help to attract star players or that may drive them away.

 It was a particularly insightful and interesting area of investigation, prompting us to delve into the real story. However, the additional research conducted to incorporate your suggestion did not align with the hypothesized outcomes. Here's the narrative:

Between 2011 and 2023, there were approximately 1,577 free agent contracts in the NBA. We sampled the top 10 largest contracts each year, resulting in 130 contracts. From this subset, we focused on unrestricted free agents, amounting to 95 players. Notably, 56 unrestricted free agents (59%) chose to re-sign with their current teams. This trend is likely influenced by the NBA's Collective Bargaining Agreement (CBA), enabling current teams to offer longer and more lucrative contracts. Hence, we can assert that contract size stands out as one of the pivotal factors in attracting star players.

Conversely, 39 players opted to join a new team. Within this group, the only instance where geographical location or a big city played a significant role in attracting a star player was LeBron James, who signed with the Los Angeles Lakers in 2018. Players like Kevin Durant (2019), Kyrie Irving (2019), and Jimmy Butler (2019) departed from their teams due to locker room issues and a desire for new challenges. Others, including Jalen Brunson (2023), Dillon Brooks (2023), and Bruce Brown (2023), left their teams seeking larger roles and more substantial contracts. Additionally, players like Kevin Durant (2016) and Steve Nash (2012) signed with new teams primarily for championship aspirations. Therefore, our findings suggest that the primary motivations for star players moving to new teams are driven by financial considerations, seeking larger roles, or pursuing championship opportunities, rather than being primarily influenced by geographical locations or the historical characteristics of a city. 

Additional Data Caveat:

The population data is sourced from macrotrends.net.

We assumed that the Brooklyn Nets and New York Knicks share the same population.

We assumed that the Los Angeles Clippers and Los Angeles Lakers share the same population.

We assumed the population for the Utah Jazz is based on the city of Salt Lake City rather than the entire state of Utah.

We used the population of Toronto for the Toronto Raptors. However, it is important to note that Toronto has the potential to represent the entire population of Canada as they are the only NBA team in the country.

Although we attempted to elucidate these considerations, we found that they may not be fitting within the current context. Consequently, these sentences are not included in the text

---

## [Editor Report · Decision Letter 1]

15 Jan 2024

Evaluating the Operational Efficiency of NBA Teams on Franchise Value: An Assessment of Data Envelopment Analysis

PONE-D-23-32235R1

Dear Dr. Lee,

We’re pleased to inform you that your manuscript has been judged scientifically suitable for publication and will be formally accepted for publication once it meets all outstanding technical requirements.

Kind regards,

Haroldo V. Ribeiro

Academic Editor

PLOS ONE

Additional Editor Comments (optional):

I have thoroughly reviewed the authors' responses to each comment provided by both reviewers and the revised version of the manuscript. All comments have been adequately addressed, eliminating the need for additional review. Furthermore, I extend my congratulations to the authors for their outstanding work. I am optimistic that the quality of their contribution will captivate the interest of PLOS ONE readers.

---

## [Editor Report · Acceptance letter]

23 Feb 2024

PONE-D-23-32235R1 

PLOS ONE

Dear Dr. Lee, 

I'm pleased to inform you that your manuscript has been deemed suitable for publication in PLOS ONE. Congratulations! Your manuscript is now being handed over to our production team.

Kind regards, 

on behalf of

Dr. Haroldo V. Ribeiro 

Academic Editor

PLOS ONE